Trophic ecology of sea urchins in coral-rocky reef systems, Ecuador

Cabanillas-Terán Nancy 1 nanchamex@gmail.com
Loor-Andrade Peggy 1
Rodríguez-Barreras Ruber 2
Cortés Jorge 3
1 Departamento Central de Investigación, Universidad Laica Eloy Alfaro de Manabí , Ciudadela Universitaria. Vía San Mateo, Manta, Manabí , Ecuador
3 Department of Biology, University of Puerto Rico at Bayamón , Puerto Rico
4 Centro de Investigación en Ciencias del Mar y Limnología (CIMAR), and Escuela de Biología, Universidad de Costa Rica , San Pedro, San José , Costa Rica
Thompson Fabiano
Electronic publication date: 2016 Jan 14
Publication date: 2016
Volume: 4
Electronic Location ID: e1578
Received 2015 Sep 10; Accepted 2015 Dec 16
Copyright: ©2016 Cabanillas-Terán et al.
Copyright year: 2016
Copyright holder: Cabanillas-Terán et al.
License: This is an open access article distributed under the terms of the Creative Commons Attribution License, which permits unrestricted use, distribution, reproduction and adaptation in any medium and for any purpose provided that it is properly attributed. For attribution, the original author(s), title, publication source (PeerJ) and either DOI or URL of the article must be cited.
License URL: https://creativecommons.org/licenses/by/4.0/

Keywords: Diadema mexicanum, Eucidaris thouarsii, Stable isotopes, Rocky reefs, Niche breadth, Eastern Tropical Pacific

Funding: Análisis de los Ecosistemas de los fondos rocosos de Ecuador continental 91740000.0000.375395 This study was funded by project: Análisis de los Ecosistemas de los fondos rocosos de Ecuador continental, # proyecto: CUP: 91740000.0000.375395 of DCI-ULEAM. The funders had no role in study design, data collection and analysis, decision to publish, or preparation of the manuscript.

==============================
Sea urchins are important grazers and influence reef development in the Eastern Tropical Pacific (ETP). Diadema mexicanum and Eucidaris thouarsii are the most important sea urchins on the Ecuadorian coastal reefs. This study provided a trophic scenario for these two species of echinoids in the coral-rocky reef bottoms of the Ecuadorian coast, using stable isotopes. We evaluated the relative proportion of algal resources assimilated, and trophic niche of the two sea urchins in the most southern coral-rocky reefs of the ETP in two sites with different disturbance level. Bayesian models were used to estimate the contribution of algal sources, niche breadth, and trophic overlap between the two species. The sea urchins behaved as opportunistic feeders, although they showed differential resource assimilation. Eucidaris thouarsii is the dominant species in disturbed environments; likewise, their niche amplitude was broader than that of D. mexicanum when conditions were not optimal. However, there was no niche overlap between the species. The Stable Isotope Analysis in R (SIAR) indicated that both sea urchins shared limiting resources in the disturbed area, mainly Dictyota spp. (contributions of up to 85% for D. mexicanum and up to 75% for E. thouarsii). The Stable Isotope Bayesian Ellipses in R (SIBER) analysis results indicated less interspecific competition in the undisturbed site. Our results suggested a trophic niche partitioning between sympatric sea urchin species in coastal areas of the ETP, but the limitation of resources could lead to trophic overlap and stronger habitat degradation.

Introduction

As a consequence of increasing human pressure, coastal ecosystems are facing a wide range of threats, such as resource exploitation and habitat modification (Wilkinson, 1999; Dumas et al., 2007; Costello et al., 2010; Rossi, 2013). Several studies have evaluated the development of rocky bottom disturbances by analyzing the densities of echinoids and the development stage of habitats (Phillips & Shima, 2006; Alvarado, Cortés & Reyes-Bonilla, 2012; Hereu et al., 2012). Some of these studies have correlated different phases of benthic substrate degradation, considering sea urchin density and their association with functional algae groups (Steneck, 1983; Steneck & Dethier, 1994). Another approach to decipher benthic dynamics is through the trophic relationships between consumers and resources using stable isotopes (Behmer & Joern, 2008). Stable isotope analysis (SIA) has been a powerful tool to study trophic ecology, especially for those species with foraging habits for which it is difficult to use traditional techniques, such as stomach content analyses. Several studies have focused on sea urchins from a stable isotope approach (e.g., Minagawa & Wada, 1984; Tomas et al., 2006; Vanderklift, Kendrick & Smith, 2006; Wing et al., 2008; Cabanillas-Terán, 2009; Rodríguez-Barreras et al., 2015).

Stable carbon and nitrogen isotope ratios provide time-integrated information regarding feeding relationships and energy flow through food webs (DeNiro & Epstein, 1981; Peterson & Fry, 1987; Vander-Zanden & Rasmussen, 2001; Carabel et al., 2006). Stable isotopes can be used to study the trophic niche of a species due to the “δ-space.” This is comparable to the n-dimensional space that ecologists refer to as a niche because an animal’s chemical composition is directly influenced by what it consumes, as well as the habitat in which it lives (Newsome et al., 2007; Parnell et al., 2010; Boecklen et al., 2011).

Carbon is a conservative tracer used to track energy sources in food webs, while nitrogen helps determine the trophic position (Minagawa & Wada, 1984; Vander-Zanden & Rasmussen, 2001; Post, 2002; Phillips, 2012; Phillips et al., 2014). Carbon (δ13C) and nitrogen (δ15N) stable isotopes have been used in marine ecosystems to determine the food habits of species (Peterson & Fry, 1987), nutrient migrations within food webs, the trophic position of organisms and their contribution at every level (Vander-Zanden & Rasmussen, 1996), the origin and transformation of the ingested organic matter (Peterson, Howarth & Garrett, 1985), or how some ecosystems have organisms that occupy similar trophic positions coexisting in high densities (Vanderklift, Kendrick & Smith, 2006). Moreover, SIA are useful to assess the ecosystem health (e.g., Cole et al., 2004; Hamaoka et al., 2010; Karube et al., 2010). For example, human influence on lake ecosystems were studied by Karube et al. (2010) and those authors found that signatures of δ13C and δ15N in macroinvertebrates of the littoral zone are indicators of anthropogenic impacts from the watershed. Inorganic nitrogen loading from the watershed was recorded in δ15N of snails.

Reef degradation currently has significant consequences for morpho-functionality of marine environments (Hoegh-Guldberg, 1999; Mumby, Foster & Fahy, 2005), and the Ecuadorian reefs are no exception (Glynn & Wellington, 1983; Glynn, 1993; Guzmán & Cortés, 1993; Glynn, 2003). Anthropogenic stressors can have synergistic effects on reefs, such as the harmful algae blooms that are becoming increasingly important drivers of variation in the sea urchin populations, as seen in other areas (Hunter & Price, 1992; Lapointe et al., 2005; Lapointe et al., 2010).

Coral-rocky reefs have rarely been studied along the Ecuadorian mainland, despite the serious threat by eutrophication, fisheries and other anthropogenic impacts (Guzmán & Cortés, 1993; Cortés, 1997; Cortés, 2011). Ecuadorian coral communities are important because they represent the southernmost distribution in the Eastern Tropical Pacific (ETP). Ecuador has no extensive reef systems, as the majority of reefs are small rocky patches with some coral colonies. Nevertheless, those areas are characterized by high biodiversity, including more than a quarter of the Ecuadorian continental fishes and a great number of echinoderms, sea fans, and scleractinian corals (Glynn et al., 2001; Glynn, 2003; Rivera & Martínez, 2011).

Sea urchins have the capability to modify the community structure through foraging, as several authors have previously mentioned (e.g., Carpenter, 1981; Carpenter, 1986; Hay, 1984; Hay & Fenical, 1988; Sala, Boudouresque & Harmelin-Vivien, 1998), and we need to elucidate what occurs in areas where there are more than one sea urchin species which dominate the substratum and their role in controlling fleshy macroalgae. The sea urchins Diadema mexicanum (Agassiz, 1863) and Eucidaris thouarsii (Agassiz & Desor, 1846) are two of the most dominant benthic grazers in the ETP (Guzmán & Cortés, 1993). These two echinoids exert a strong influence on the community structure (Lawrence, 1975; Glynn, Wellington & Birkeland, 1979; Andrew, 1989; Underwood, 1992). In the ETP, the sea urchin E. thouarsii could be described as a major herbivore in rocky reef bottoms. Its preferential resource appeared to be benthic algal turf and macroalgae, but if those were not available, it feeds on other organisms, such as the corals Pavona clavus, Pocillopora spp. and Porites lobata (Glynn, Wellington & Birkeland, 1979; Glynn & Wellington, 1983; Reaka-Kudla, Feingold & Glynn, 1996).

The ratios of δ15N and δ13C in consumers are strongly influenced by their food resources (Phillips et al., 2014) and it is necessary to identify their ecological role, not only by their capacity to structure the environment, but to understand the dynamics of coexistence of the sea urchin populations along the Ecuadorian coast. The relative position of δ13C vs. δ15N echinoids species can be displayed in a bi-plot and help to understand food web structure and organism responses to niche shifts, diet variability and human impact (Layman et al., 2007). The aim of this study was to improve the knowledge and understanding of the trophic biology of D. mexicanum and E. thouarsii in Ecuadorian rocky reefs. We determined the stable isotopes of carbon and nitrogen isotope for both sea urchin species. The complexity of the littoral zone was analyzed using stable isotopes to understand the trophic interactions of these two echinoids in areas with different degree of human impact. We assume that the more developed rocky-reef and substrate associated to coral coverage will favor habitats with complex trophic interactions (see Duffy et al., 2007; Alvarez-Filip et al., 2009; Graham & Nash, 2013), resulting in wider isotopic echinoids niche breadth.

Material & Methods

Site descriptions

This study was conducted between May to September of 2013, at two localities, Los Ahorcados (LA: 01°40′42″S; 80°51′58″W) and Perpetuo Socorro (PS: 0°55′40″S; 80°44′25″W), in Manabí province, Ecuador (Fig. 1). The LA site was a small group of rocky islets located near Machalilla National Park. Although this area was not considered a protected area, it had a very high diversity of scleractinians and octocorals (Rivera & Martínez, 2011). LA presented a rocky bottom with clear geomorphologic differences between the leeward and windward areas. The leeward area were 25 m depth, and the windward side was mainly build by octocorals (22 species), and hexacorals, such as Pavona spp., the branching corals Pocillopora spp. and solitary corals. The PS site is located in front of the Port of Manta (1.5 km), one of the most important ports in Ecuador for large pelagic fisheries (Villón & Beltrán, 1998a; Villón & Beltrán, 1998b; Martínez-Ortiz et al., 2007) and greatly impacted by anthropogenic activities (see details in Table 1). PS rocky reef is a homogenous bottom of 7–9 m depth, and had a substrate consisted mainly of a mixture of rock and sand with scarce scleractinian corals (Pavona spp. and Pocillopora spp.) and gorgonians (mainly Leptogorgia alba).

In order to distinguish both sites, a coral-rocky reef category was used for this study, which was developed taking into account habitat complexity and type of disturbance to establish two categories, namely disturbed and undisturbed (Table 1). The distance of human impact to the sites, size of the fleet and type of human disturbance were considered. The rugosity index (RI), which is the ratio of a length of chain following the reef contour to the linear distance between its start and end point (modified of Risk, 1972) was used. To calculate the RI, we used a three-meter chain five times equitably-distributed along 15 transects (20 m). The average RI obtained with transects was used to determine the rugosity level per site, where larger numbers indicate higher complexity following Alvarez-Filip et al. (2009) and Alvarez-Filip et al. (2011). Therefore, values of RI < 1.5 were considered low complexity and RI > 1.5 were defined as complex.

Figure 1 Study area and sampling sites in the coast of Ecuador: Los Ahorcados (LA) and Perpetuo Socorro (PS).

Table 1 Category of coral-rocky reef sites and source of human impact.

Site	Source of human impact	Human population density (ind km−2)	Distance from sampling site to Source of human impact (km)	Rugosity index (RI)	Category	
Los Ahorcados (LA)	Artesanal fishery + hotel zone	54.55	17.24	2.32	Undisturbed	
Perpetuo Socorro (PS)	Artesanal and industrial Fishery + hotel zone + industry discharge	1046.34	3.43	1.10	Disturbed	

Collecting and processing data

We collected algal samples for identification, to calculate biomass, and to carry out SIA. Algal biomass was measured using twelve 50 × 50 cm quadrats per site. The quadrats were located randomly within the sea urchin habitat. The substrate inside each quadrat was scrapped, carefully removed, collected in bags, and frozen for later analysis. Macroalgae were identified to the lowest possible taxonomic level using the available keys (Abbott & Hollenberg, 1976; Afonso-Carrillo & Sansón, 1999; Littler & Littler, 2010). The sampled invertebrate and algal species for this study are not threatened. The necessary permits were obtained from the Ministry of Environment of Ecuador (014AT-DPAM-MAE).

In areas where the algal cover was dominated mainly by turf species (following the morpho-functional category of Guidetti, 2006), we used a sniffer with a dense mesh bag coupled to a compressed air tank. In the laboratory, individuals were separated into species and gently washed with distilled water and dried in an oven at 50 °C for 24 h to measure the dry weight.

We collected four individuals of D. mexicanum and six of E. thouarsii in LA and twelve individuals of D. mexicanum and eight of E. thouarsii in PS at the same depth range (8–10 m). Only individuals greater than 5.0 cm in test diameter were collected to avoid any effect of the development stage. The samples were frozen shortly after collection and processed at the laboratory. The muscles of Aristotle’s lanterns were removed carefully and washed from the stomach contents to estimate algal assimilation by D. mexicanum and E. thouarsii. This tissue provides a time-integrated measure of assimilated sources (e.g., Michener & Schell, 1994; Ben-David & Schell, 2001; Polunin et al., 2001; Phillips & Koch, 2002; Rodríguez, 2003; Tomas et al., 2006).

The algal and echinoids muscle samples were rinsed with filtered water, dried at 50 °C during 36 h, ground to a fine powder and placed in glass vial for isotope analyses. To remove carbonates from some algal species (Lobophora variegata and Polysiphonia spp.), the samples were washed with diluted HCl at 1 N prior to drying to avoid disturbance in the mass spectrometer reading. A subsample was taken of each alga and muscle (∾1 mg) to evaluate the 13C/12C and 15N∕14N ratios using a Thermo Electron Delta V Advantage Mass Spectrometer. Carbon and nitrogen samples were analyzed in a dual isotope mode at the Geology Department, University of Florida, Gainesville, Florida.

The isotope samples were loaded into Eppendorf capsules and placed in a 50-position automated Zero Blank sample carousel on a Carlo Erba NA1500 CNS elemental analyzer. After combustion in a quartz column at 1,020 °C in an oxygen-rich atmosphere, the sample gas was transported in a He carrier stream and passed through a hot reduction column (650 °C) consisting of elemental copper to remove oxygen. The effluent stream then passed through a chemical (magnesium perchlorate) trap to remove water, followed by a 3 m Gas chromatography (GC) column at 45 °C to separate N2 from CO2. The sample gas next passed into a ConFlo II preparation system and into the inlet of a mass spectrometer running in continuous flow mode, where the sample gas was measured relative to laboratory reference N2 and CO2 gases. The carbon isotopic results were expressed in standard delta notation relative to Vienna Pee Dee Belemnite (VPDB). The nitrogen isotopic results were expressed in standard delta notation relative to atmospheric air. The standard deviations of δ13C and δ15N replicate analyses were estimated; the precision values were 0.074 and 0.148 for carbon and nitrogen isotope measurements, respectively. Carbon and nitrogen samples were analyzed in a dual isotope mode. Ratios are expressed as: δX ‰=Rsample∕Rstandard−1×1,000;whereX=13Cor15NandRsample=13C∕12Cor15N∕14N.

Data analysis

The relative contribution of algae to the diet of the sea urchins D. mexicanum and E. thouarsii was estimated with a Bayesian isotopic mixing model (SIAR, Parnell & Jackson, 2013), which included the isotopic signatures, fractionation and variability to estimate the probability distribution of the contribution of the food source to a mixture. This procedure supplied accurate information about the contribution of algal species to the sea urchin tissues recognized the main components of the diet under different conditions (Peterson, 1999; Fry, 2006; Wing et al., 2008). Lipid extraction in sea urchins was not necessary since Aristotle lantern’s muscle is low in lipids, on the other hand when the C:N ratios are lower than 3.5 it is not recommended (Post et al., 2007) see Table S1. The isotopic discrimination factor values used to run the model were 2.4 ± 1.6‰ (mean ± SD) for δ15N, and 0.4 ± 1.3‰ for δ13C (Fry & Sherr, 1984; Minagawa & Wada, 1984; Michener & Schell, 1994; Moore & Semmens, 2008). The results of the mixing model showing the calculated sea urchin dietary proportions were represented as box plots, indicating the 25%, 75%, and 95% of credibility intervals (Fig. 2).

Figure 2 Contribution rates of algae to the diet of the two sea urchin species.

Results are shown as 25, 75 and 95% of credibility intervals. (A) Represents the contribution for Diadema mexicanum in Los Ahorcados (LA), (B) for Eucidaris thouarsii in LA, (C) D. mexicanumin Perpetuo Socorro (PS), and (D) E. thouarsii in PS.

The niche width and overlap for the sea urchins were estimated with Stable Isotope Bayesian Ellipses in R (SIBER) (Jackson et al., 2011) from the SIAR package (Parnell & Jackson, 2013). This analysis uses metrics based on ellipses and provides the standard ellipse corrected area (SEAc) used as the trophic niche breadth and the overlap between ellipses, where values close to 1 represent a higher trophic overlap.

Prior to the statistical analysis, the homogeneity and normality of variance were tested by performing a Kolmogorov–Smirnov and a Cochran’s test (Zar, 2010). Statistical difference was performed comparing δ15N and δ13C values between species. In addition, the algal biomass between sites was evaluated with a one-way ANOVA, with site as a fixed factor. The statistical analyses were performed using R with an alpha of 0.05 (R Core Team, 2014).

Results

The benthic communities in Ecuadorian rocky reefs ranged between habitats dominated by macroalgae and live corals (LA), and habitats dominated by turf and coral skeletons (PS). Site estimates for perturbation and complexity are outlined in Table 1. LA is a site with structural complexity and dominance of branched erect algae, while PS has low structural complexity and dominance of turf (Table 1).

The algae collected in LA were Asparagopsis armata, Dictyota dichotoma, Lobophora variegata, Polysiphonia spp., and Sargassum spp., while in PS were D. dichotoma, L. variegata, and Polysiphonia spp. The greatest algal biomass was recorded for L. variegata at both sites, while D. dichotoma was the algae with the lowest biomass at both localities (Table 2). Overall, the biomass average values ranged from 35.8 ± 9.73 g (dry weight) m−2 for PS to 143.00 ± 20.67 g m−2 in LA. We found significant differences between both sites (ANOVA, df = 1, F = 3.60, p < 0.01). The overall algal δ15N fluctuated from 5.05 to 9.49‰ (Table 3). PS displayed the highest mean values of nitrogen with D. dichotoma (7.60 ± 0.53‰). At LA, Polysiphonia spp. exhibited the highest mean value for nitrogen (7.19 ± 1.13‰). We found significant differences in δ15N between sites (ANOVA, df = 1, F = 5.29, p = 0.02), taking into account all the algae isotopic signatures. As for δ13C, ratios fluctuated from −23.65 to −6.90‰, with LA displaying the most negative values (A. armata). There was no significant difference in δ13C among sites (ANOVA, DF = 1, F = 1.41, p > 0.05).

Table 2 Average algal biomass in grams (dry weight) m−2 ± standard deviation at Los Ahorcados (LA) and Perpetuo Socorro (PS)

Species	LA	PS	
A. armata	34.32 ± 16.98	–	
D. dichotoma	4.69 ± 1.90	0.60 ± 0.20	
L. variegata	66.77 ± 24.52	23.26 ± 12.61	
Polysiphonia spp.	30.73 ± 12.82	16.38 ± 6.26	
Sargassum spp.	5.94 ± 3.09	–	

Table 3 Mean ± standard deviation values of δ13C and δ15N of algal genus considered in the mixing model analysis taken from Los Ahorcados and Perpetuo Socorro.

	Los Ahorcados		Perpetuo Socorro	
Species	δ13C	δ15N	Species	δ13C	δ15N	
A. armata (n = 4)	−23.63 ± 0.10	5.68 ± 0.02	–	–	–	
D. dichotomaa (n = 4)	−17.30 ± 1.94	6.65 ± 0.791	D. dichotoma (n = 3)	−15.27 ± 3.05	7.60 ± 0.53	
L. variegata (n = 4)	−15.73 ± 3.331	5.89 ± 0.638	L. variegata (n = 3)	−12.02 ± 0.60	7.06 ± 1.08	
Polysiphonia spp. (n = 6)	−9.33 ± 1.759	7.19 ± 1.129	Polysiphonia spp. (n = 4)	−14.72 ± 3.04	7.38 ± 0.36	
Sargassum spp. (n = 4)	−18.30 ± 0.07	6.97 ± 0.06	–	–	

Values of δ15N were particularly different between the two species of sea urchins (ANOVA, df = 1, F = 20.10, p < 0.001). The isotopic value of δ15N for D. mexicanum ranged from 11.38 to 12.99‰, whereas E. thouarsii displayed values from 12.31 to 14.15‰. The average values of δ13C and δ15N estimated for D. mexicanum in LA were −16.67 ± 0.04 and 11.53 ± 0.14‰, respectively, while E. thouarsii displayed −15.46 ± 0.16‰ and 12.84 ± 0.40‰, respectively. In PS, the D. mexicanum isotopic signals were −16.25 ± 0.39‰ for δ13C and 12.62 ± 0.22‰ for δ15N; while E. thouarsii displayed −15.41 ± 0.43‰ for δ13C and 13.54 ± 0.47‰ for δ15N. We found significant differences in δ13C between species (ANOVA, df 1, F = 49.31, p < 0.0001), and the most negative values were found at LA. The δ15N showed the same patterns as those algae (higher values for PS). δ15N ratios of both sea urchins differed between the study sites, as LA reported lower values than PS (ANOVA, df 1, F = 7.59, p < 0.01). The most notorious difference was due to D. mexicanum (ANOVA, df 1, F = 82.41, p < 0.0001).

The mixing models provided evidence for the contribution of different algal resources for both sites and species. The SIAR analysis showed that Sargassum spp. was the most important resource for D. mexicanum in LA (up to 43%), followed by D. dichotoma and A. armata as secondary resources (up to 37% for both). Likewise, Sargassum spp. was the main algal resource for E. thouarsii in the same locality (up to 44%), followed by Polysiphonia spp. (up to 41%) (Table 4). Contrasting, at PS the main macroalgal contributor was D. dichotoma for both sea urchins (Fig. 2), with up to 85% of the proportional contribution for D. mexicanum and close to 75% for E. thouarsii.Table 5 shows data on isotopic niche breadth as measured by the corrected standard ellipse area (SEAc). The main difference in the trophic niche breadth was caused by E. thouarsii with a difference probability of 52%; overlap between species isotopic niches was not found in any case (Fig. 3), but the SEAc was higher for E. thouarsii in both sites with 0.25 in LA and 0.46 in PS (Table 5).

Table 4 Average percentage (%) contribution of algal species to the diet of the sea urchins D. mexicanum and E. thouarsii at Los Ahorcados (LA) and Perpetuo Socorro (PS) produced by the SIAR model using isotope values from algae.

Minimum and maximum values for each algae are shown in parentheses.

	Diadema mexicanum	Eucidaris thouarsii	
Species	LA	PS	LA	PS	
A. armata	21 (2–37)	–	14 (0–28)	–	
D. dichotoma	20 (0–37)	52 (21–85)	19 (0–37)	44 (13–75)	
L. variegata	16 (0–32)	9 (0–23)	15 (0–31)	19 (0–38)	
Polysiphonia spp.	20 (04–35)	38 (3–67)	28 (15–41)	38 (4–66)	
Sargassum spp.	23 (1–43)	–	24 (1–44)	–	

Table 5 Trophic niche breadth of sea urchins calculated by SIBER analysis of muscle values.

SEAc: corrected standard ellipse area. The right column shows statistical differences in SEA.

Species	SEAc	Ellipses areas: group differences probability (%)	
D. mexicanum (LA)	0.005	1 vs. 2 (10.4)a	
D. mexicanum (PS)	0.218	
E. thouarsii (LA)	0.250	1 vs. 2 (52.0)a	
E. thouarsii (PS)	0.457	
Notes.

a Group 1: Los Ahorcados (LA); Group 2: Perpetuo Socorro (PS).

Figure 3 Isotope niche breadth of the echinoids, D. mexicanum (circles) and E. thouarsii (triangles) in Los Ahorcados (white symbols and solid line) and Perpetuo Socorro (black symbols and dotted line).

Discussion

There is very little information on the ecology of the Ecuadorian coast, and no data pertaining to trophic relationships among sea urchins, apart from this study. The majority of the available information on Ecuador came from studies conducted on Galapagos reefs (Glynn, Wellington & Birkeland, 1979; Glynn & Wellington, 1983; Glynn, 2003; Glynn, 2004; Glynn et al., 2009). The rocky reefs examined in this study were selected to establish the baseline of the trophic ecology of two rocky reef areas, with different disturbance levels in the Ecuadorian mainland coast. The presence of D. mexicanum was related to the rocky bottoms of LA, where algal presence were more frequent than in the disturbed site (PS). The population density of E. thouarsii was higher at the disturbed site (N Cabanillas-Terán, 2013, unpublished data). This study demonstrated that algal abundance is not always equivalent to assimilation by the consumer. For instance, L. variegata displayed the lowest dietary contribution at PS and LA for both sea urchins, although it exhibited the highest average biomass at both sites. Grazing preference by D. mexicanum and E. thouarsii was not related to algal biomass.

The isotopic results characterized different algal assemblages that were specific to each rocky reef bottom (branched macroalgae for LA and turf for PS). The values of δ15N in algae in this study ranged from 5.05‰ to 9.49‰. This result agreed with the ranges of variation reported in other studies (Owens, 1987). The values of δ13C fluctuated from −23.65 to −6.90‰ and agreed with data from Fry & Sherr (1984), who reviewed the δ13C data of benthic algae, noting that values ranged between −30 and −5‰. The different algae species constituting the community of LA showed isotopic values that were closer together, but with a broader cloud distribution in the C vs. N biplot of points relative to what was observed in PS. This suggests a more complex trophic net and shows how primary consumers interact with their resources (McClanahan, 1988; Phillips & Gregg, 2003).

The isotopic ratios of δ15N could be influenced by two main factors. One factor pertains to changes in dissolved nitrogen, although these changes primarily affect the microscopic algal communities or communities living near upwelling zones (Jennings et al., 1997; Polunin & Pinnegar, 2002). The other factor is the anthropogenic impact (Bode, Alvarez-Ossorio & Varela, 2006), affecting the communities near the coastline. In this case, the community most affected by urban impact was PS, located in front of Manta Port. In this port, human density is higher than 1,000 ind/km−2, and artisanal and industrial fishery contribute to nitrogen input, as well as the hotel zone and discharges from tuna processing.

For algae found in both sites (D. dichotoma, L. variegata and Polysiphonia spp.), the average δ15N were higher in PS. This agrees with other areas with high anthropogenic influence where δ15N values tended to be higher (Wada, Kadonaga & Matsuo, 1975; Michener & Schell, 1994). Although both localities shared species, the isotopic values for both localities were different because each system had its own structure. The erected branched algae A. armata and Sargassum spp. (not found in PS), contributed to the structural complexity founded in LA.

Variations in carbon and nitrogen ratios gave us information on trophic spectrum inherent to each site and the contribution of algal species to the sea urchin tissues display information about how consumers assimilate the resources when they inhabit disturbed and/or undisturbed sites. Although both sea urchin species can share the same food resources, we found that their ecological roles were different and there are differences between species in terms of assimilation. This could explain the fact that δ15N values in the muscle of E. thouarsii were higher for both localities, even though both sea urchin species showed a preference for the same species D. dichotoma. No overlap of isotope niche breadth of the echinoids was found between the two species (Fig. 3), but the isotopic values between species at PS were closer, suggesting increased competition due to the lack of resources. This result coincided with the mixing model because the two species of sea urchins preferentially consumed similar proportions of the same species. Moreover the SEAc was larger for E. thouarsii at both sites, and in LA the niche trophic distance between D. mexicanum and E. thouarsii was very conspicuous, while in PS the two species of sea urchins are closer (Fig. 3). A low degree of feeding specialization suggests that the sea urchins adapt their foraging behavior to algae availability, being most evident for E. thouarsii, that exhibits a broader trophic niche.

The grazing behavior of these sea urchins agreed with the findings by Glynn, Wellington & Birkeland (1979) in the Galápagos Islands, as their grazing was stronger in those areas with 30% or less coral cover. Previous studies highlighted that E. thouarsii limited coral growth, as this echinoid interfered with the development of the reef frame and with the ability to modify the habitat structure (Bak, 1994; Carpenter, 1981; Sonnenholzner, Ladah & Lafferty, 2009). We considered D. mexicanum to be an important grazer for the rocky bottoms ecosystems, considering that changes in its population caused significant changes in the algal cover of those areas.

Our results supported the evidence that D. mexicanum and E. thouarsii were coexistent species that play a significant role as herbivores. Nevertheless, they apparently eat whatever they find, and the food items are incorporated differentially between the species. Diadema mexicanum grazing effect on algal diversity and community structure is important in the process of formation and maintenance of rocky reefs in Ecuador. This has also been observed in other areas of the ETP where D. mexicanum has a relevant role in the recruitment of corals (Alvarado, Cortés & Reyes-Bonilla, 2012). This was also observed in Caribbean reefs with D. antillarum (Macintyre, Glynn & Hinds, 2005; Mumby et al., 2006; Idjadi, Haring & Precht, 2010; Sandin & McNamara, 2012), and in shaping the sublittoral ecosystems of the Canary Islands with D. africanum (Alves et al., 2003; Tuya et al., 2004; Hernández et al., 2005; Hernández et al., 2008; Sangil et al., 2014; Cabanillas-Terán et al., 2015).

The standard ellipses areas values (Table 5) indicated that niche partitioning may vary depending on different disturbance levels between sites; however, the diets of D. mexicanum and E. thouarsii not only depend on the disturbance condition. For instance, Dictyota dichotoma was an important component of the diet of D. mexicanum and E. thouarsii in the disturbed and undisturbed sites, while Polysiphonia spp. was important in disturbed bottoms, where isotopic algal signals are closer to each other. This could lead to a greater number of resource overlap at PS than at LA.

Differential assimilation and niche partitioning are just snapshots. It is important to depict how the shape of the food web varies in time and space (Layman et al., 2007; Schmidt et al., 2007), so it is necessary to carry out more extensive spatial and temporal research. Likewise, it is necessary to deepen research to analyze if the narrower niche amplitude (SEAc) of D. mexicanum and its associated presence to scleractian corals (at LA) is consistent to what is happening in the Caribbean, where its presence provides suitable habitat for coral recruitment. The feeding success of herbivores is associated with the competition level for resources; therefore, sympatric species are exposed to a potential trophic overlap. The most pristine zone (LA) exhibited smaller SEAc (considering values per species) and nitrogen values, which indicate a trophic niche partitioning between the main sea urchins on the Ecuadorian coast. However, the limitation of resources could lead to trophic overlap and stronger habitat degradation.

Supplemental Information

Table S1 C:N ratio

Mean ±standard deviation values of C:N ratios of algal species and sea urchins taken from of Los Ahorcados (LA) and Perpetuo Socorro (PS).

Click here for additional data file.

Data S1 Sea urchins density dataset

Click here for additional data file.

The first author is grateful to Limber Alcivar and Jorge Figueroa for their scientific diving, logistic help during fieldwork and laboratory work. Thanks to the Departamento Central de Investigación-Universidad Laica Eloy Alfaro de Manabí. We want to acknowledge Norman Mercado and José Luis Varela for their valuable comments.

Additional Information and Declarations

Competing Interests

Author Contributions

Field Study Permissions

Data Availability

The authors declare there are no competing insterest.

Nancy Cabanillas-Terán conceived and designed the experiments, performed the experiments, analyzed the data, contributed reagents/materials/analysis tools, wrote the paper, prepared figures and/or tables, reviewed drafts of the paper.

Peggy Loor-Andrade, Ruber Rodríguez-Barreras and Jorge Cortés analyzed the data, wrote the paper, prepared figures and/or tables, reviewed drafts of the paper.

The following information was supplied relating to field study approvals (i.e., approving body and any reference numbers):

Ministerio de Medio Ambiente de Ecuador

014AT-DPAM-MAE.

The following information was supplied regarding data availability:

Data S1.

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
