# Peer review of "Trophic ecology of sea urchins in coral-rocky reef systems, Ecuador"

_PeerJ, doi:10.7717/peerj.1578_

## Round 0.1 · original submission · Major Revisions

Dear Authors:

Consider the inconsistencies, particularly the one raised by ref#2.

Reviewer 1 ·

Basic reporting

The study is original and brings relevant results, figures have good quality, but the text must be improved, especially the sessions “materials and methods”, “results” and “discussion”. The authors may improve the manuscript following a sequence of clear ideas, avoiding long sentences, avoiding mixing subjects, writing short and objective sentences. It was difficult to evaluate the scientific contribution because the structure of the text has low quality and is not clear. I suggested changes throughout the text and I highlighted parts which most need revision, but I strongly recommend a professional revision from a native speaker of English. The quality of the text must defintely be improved before publication. There is an error in Table 3 in the values of nitrogen. Raw data available in excel were not shown in the results and were not properly discussed.

Experimental design

"No Comments"

Validity of the findings

"No Comments"

Annotated reviews are not available for download in order to protect the identity of reviewers who chose to remain anonymous.

Reviewer 2 ·

Basic reporting

I believe the paper addresses a current gap regarding trophic relationships of sea urchins from Ecuador. I suggest several changes in the introduction and material and methods sections. Some relevant data are missing and this problem should be considered by the authors. After the inclusion of these new data, the discussion should also be improved. Overall, after these modifications the paper will be ready for publication. More details can be seen on the attached file.

Experimental design

With the C/N ratio, it will be possible to determine if lipids extraction is relevant or not to the current dataset. Also, a statement regarding what was defined in this paper (lines 115-117) do not correspond to the methods chosen. More details in the attached document. Please check this incoherence.

Validity of the findings

It's important that the authors provide C/N ratio and metric data. This will improve the paper and the understanding of the readers. Please rewrite the last paragraph from your introduction, since you didn't determine trophic levels. A trophic level calculation includes an enrichment factor and a baseline, going beyond a mean 15N isotope value for a species (please check Rasmussen, 2001 - Limnl. Oceanogr. 46(8), 2061-2066; or Fisket al, 2001 - Environ. Sci. Technol. 35, 732-738 for details). Finally, rewrite your final statements in the discussion based on your original hypothesis.

Additional comments

Please include the C/N ratio and the metric data. They're very important to your paper. Also, discuss, or make it clear, the possible contribution of different preys (not only algae) to the feeding of the sea urchins investigated. Explore if the absence of such preys limit or not your findings. I think it would be interesting to calculate the trophic levels and, finally, rewrite your discussion in light of these data. I believe these modifications will improve your paper.

Annotated reviews are not available for download in order to protect the identity of reviewers who chose to remain anonymous.

---

## Round 0.2 · accepted · Accept

Congratulations for the accepted MS.